# Learnable Tree Filter for Structure-preserving Feature Transform

**Lin Song**[1*]  **Yanwei Li**[2,3*]  **Zeming Li**[4]  **Gang Yu**[4]  **Hongbin Sun**[1†]
**Jian Sun**[4]  **Nanning Zheng**[1]
[1] Institute of Artificial Intelligence and Robotics, Xi'an Jiaotong Univeristy.
[2] Institute of Automation, Chinese Academy of Sciences.
[3] University of Chinese Academy of Sciences. [4] Megvii Inc. (Face++).
stevengrove@stu.xjtu.edu.cn, liyanwei2017@ia.ac.cn,
{hsun, nnzheng}@mail.xjtu.edu.cn, {lizeming, yugang, sunjian}@megvii.com

## Abstract

Learning discriminative global features plays a vital role in semantic segmentation. And most of the existing methods adopt stacks of local convolutions or non-local blocks to capture long-range context. However, due to the absence of spatial structure preservation, these operators ignore the object details when enlarging receptive fields. In this paper, we propose the *learnable tree filter* to form a generic *tree filtering module* that leverages the structural property of minimal spanning tree to model long-range dependencies while preserving the details. Furthermore, we propose a highly efficient linear-time algorithm to reduce resource consumption. Thus, the designed modules can be plugged into existing deep neural networks conveniently. To this end, tree filtering modules are embedded to formulate a unified framework for semantic segmentation. We conduct extensive ablation studies to elaborate on the effectiveness and efficiency of the proposed method. Specifically, it attains better performance with much less overhead compared with the classic PSP block and Non-local operation under the same backbone. Our approach is proved to achieve consistent improvements on several benchmarks without bells-and-whistles. Code and models are available at https://github.com/StevenGrove/TreeFilter-Torch.

## 1  Introduction

Scene perception, based on semantic segmentation, is a fundamental yet challenging topic in the vision field. The goal is to assign each pixel in the image with one of several predefined categories. With the developments of convolutional neural networks (CNN), it has achieved promising results using improved feature representations. Recently, numerous approaches have been proposed to capture larger receptive fields for global context aggregation [1–5], which can be divided into *local* and *non-local* solutions according to their pipelines.

Traditional *local* approaches enlarge receptive fields by stacking conventional convolutions [6–8] or their variants (*e.g.,* atrous convolutions [9, 2]). Moreover, the distribution of impact within a receptive field in deep stacks of convolutions converges to Gaussian [10], without detailed structure preservation (the pertinent details, which are proved to be effective in feature representation [11, 12]). Considering the limitation of local operations, several *non-local* solutions have been proposed to model *long-range* feature dependencies directly, such as convolutional methods (*e.g.*, non-local operations [13], PSP [3] and ASPP modules [2, 14, 15]) and graph-based neural networks [16–18]. However, due to the absence of structure-preserving property, which considers both spatial

---

[*]Equal contribution. This work was done in Megvii Research.
[†]Corresponding author.

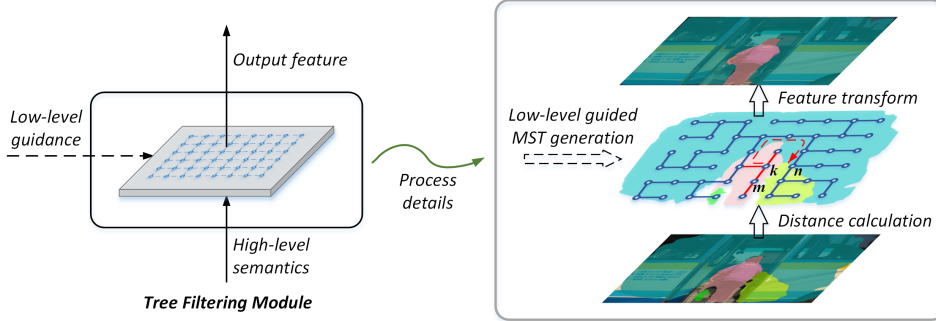

Figure 1: Toy illustration of the tree filtering module. Given a detail-rich feature map from low-level stage, we first measure the dissimilarity between each pixel and its' *quad* neighbours. Then, the MST is built upon the *4-connected* planar graph to formulate a *learnable tree filter*. The edge between two vertices denotes the distance calculated from high-level semantics. Red edges indicate the close relation with vertex $k$. The intra-class inconsistency could be alleviated after feature transform.

distance and feature dissimilarity, the object details are still neglected. Going one step further, the abovementioned operations can be viewed as *coarse* feature aggregation methods, which means they fail to explicitly preserve the original structures when capturing long-range context cues.

In this work, we aim to fix this issue by introducing a novel network component that enables efficient structure-preserving feature transform, called *learnable tree filter*. Motivated by traditional tree filter [19], a widely used image denoising operator, we utilize *tree-structured graphs* to model long-range dependencies while preserving the object structure. To this end, we first build the *low-level guided* minimum spanning trees (MST), as illustrated in Fig. 1. Then the distance between vertices in MST are calculated based on the high-level semantics, which can be optimized in backpropagation. For example, the dissimilarity $w_{k,m}$ between vertex $k$ and $m$ in Fig. 1 is calculated from semantic-rich feature embeddings. Thus, combined with the structural property of MST, the spatial distance and feature dissimilarity have been modeled into tree-structured graph simultaneously (*e.g.,* the distance between vertex $k$ and its spatially adjacent one *n* has been enlarged in Fig. 1, for that more edges with dissimilarities are calculated when approaching *n*). To enable the potential for practical application, we further propose an efficient algorithm which reduces the $\mathcal{O}(N^2)$ complexity of brute force implementation to linear-time consumption. Consequently, different from conditional random fields (CRF) [20–22], the formulated modules can be embedded into several neural network layers for differentiable optimization.

In principle, the proposed *tree filtering module* is fundamentally different from most CNN based methods. The approach exploits a new dimension: *tree-structure graph is utilized for structure-preserving feature transform, bring detailed object structure as well as long-range dependencies*. With the designed efficient implementation, the proposed approach can be applied for multi-scale feature aggregation with much less resource consumption. Moreover, extensive ablation studies have been conducted to elaborate on its superiority in both performance and efficiency even comparing with PSP block [3] and Non-local block [13]. Experiments on two well-known datasets (PASCAL VOC 2012 [23] and Cityscapes [24]) also prove the effectiveness of the proposed method.

## 2 Learnable Tree Filter

To preserve object structures when capturing long-range dependencies, we formulate the proposed *learnable tree filter* into a generic feature extractor, called *tree filtering module*. Thus, it can be easily embedded in deep neural networks for end-to-end optimization. In this section, we firstly introduce the learnable tree filtering operator. And then the efficient implementation is presented for practical applications. The constructed framework for semantic segmentation is elaborated at last.

### 2.1 Formulation

First, we represent the *low-level* feature as a undirected graph $G = (V, E)$, with the dissimilarity weight $\omega$ for edges. The vertices $V$ are the semantic features, and the interconnections of them can

be denoted as $E$. Low-level stage feature map, which contains abundant object details, is adopted as the guidance for *4-connected planar graph* construction, as illustrated in Fig.1. Thus, a spanning tree can be generated from the graph by performing a pruning algorithm to remove the edges with the substantial dissimilarity. From this perspective, the graph $G$ turns out to be the *minimum spanning tree* (MST) whose sum of dissimilarity weights is minimum out of all spanning trees. The property of MST ensures preferential interaction among similar vertices. Motivated by traditional tree filter [19], a generic tree filtering module in the deep neural network can be formulated as:

$$\boldsymbol{y}_i = \frac{1}{z_i} \sum_{\forall j \in \Omega} S\left(\boldsymbol{E}_{i,j}\right) f\left(\boldsymbol{x}_j\right). \tag{1}$$

Where $i$ and $j$ indicate the index of vertices, $\Omega$ denotes the set of all vertices in the tree $G$, $\boldsymbol{x}$ represents the input encoded features and $\boldsymbol{y}$ means the output signal sharing the same shape with $\boldsymbol{x}$. $\boldsymbol{E}_{i,j}$ is a hyperedge which contains a set of vertices traced from vertex $i$ to $j$ in $G$. The similarity function $S$ projects the features of the hyperedge into a positive scalar value, as described in Eq. 2. The unary function $f(\cdot)$ represents the feature embedding transformation. $z_i$ is the summation of similarity $S(\boldsymbol{E}_{i,j})$ alone with $j$ to normalize the response.

$$S\left(\boldsymbol{E}_{i,j}\right) = \exp\left(-D\left(i,j\right)\right), \quad \text{where } D\left(i,j\right) = D\left(j,i\right) = \sum_{(k,m) \in \boldsymbol{E}_{i,j}} \boldsymbol{\omega}_{k,m}. \tag{2}$$

According to the formula in Eq. 1, the tree filtering operation can be considered as one kind of *weighted-average filter*. The variable $\boldsymbol{\omega}_{k,m}$ indicates the dissimilarity between adjacent vertices ($k$ and $m$) that can be computed by a pairwise function (here we adopt *Euclidean distance*). The distance $D$ between two vertices ($i$ and $j$) is defined as summation of dissimilarity $\boldsymbol{\omega}_{k,m}$ along the path in hyperedge $\boldsymbol{E}_{i,j}$. Note that $D$ degenerates into spatial distance when $\boldsymbol{\omega}$ is set to a constant matrix. Since $\boldsymbol{\omega}$ actually measures pairwise distance in the embedded space, the aggregation along the pre-generated tree considers spatial distance and feature difference simultaneously.

$$\boldsymbol{y}_i = \frac{1}{z_i} \sum_{\forall j \in \Omega} f\left(\boldsymbol{x}_j\right) \prod_{(k,m) \in \boldsymbol{E}_{i,j}} \exp\left(-\boldsymbol{\omega}_{k,m}\right), \quad \text{where } z_i = \sum_{\forall j \in \Omega} \prod_{(k,m) \in \boldsymbol{E}_{i,j}} \exp\left(-\boldsymbol{\omega}_{k,m}\right). \tag{3}$$

The tree filtering module can be reformulated to Eq. 3. Obviously, the input feature $\boldsymbol{x}_j$ and dissimilarity $\boldsymbol{\omega}_{k,m}$ take responsibility for the output response $\boldsymbol{y}_i$. Therefore, the derivative of output with respect to input variables can be derived as Eq. 4 and Eq. 5. $\boldsymbol{V}_m^i$ in Eq. 5 is defined with the children of vertex $m$ in the tree whose root node is the vertex $i$.

$$\frac{\partial \boldsymbol{y}_i}{\partial \boldsymbol{x}_j} = \frac{S\left(\boldsymbol{E}_{i,j}\right)}{z_i} \frac{\partial f\left(\boldsymbol{x}_j\right)}{\partial \boldsymbol{x}_j}, \tag{4}$$

$$\frac{\partial \boldsymbol{y}_i}{\partial \boldsymbol{\omega}_{k,m}} = \frac{S\left(\boldsymbol{E}_{i,k}\right)}{z_i} \frac{\partial S\left(\boldsymbol{E}_{k,m}\right)}{\partial \boldsymbol{\omega}_{k,m}} \left( \sum_{j \in \boldsymbol{V}_m^i} S\left(\boldsymbol{E}_{m,j}\right) f\left(\boldsymbol{x}_j\right) - \boldsymbol{y}_i z_m \right). \tag{5}$$

In this way, the proposed tree filtering operator can be formulated as a *differentiable* module, which can be optimized by the backpropagation algorithm in an end-to-end manner.

## 2.2 Efficient Implementation

Let $N$ denotes the number of vertices in the tree $G$. The tree filtering module needs to be accumulated $N$ times for each output vertex. For each channel, the computational complexity of brute force implementation is $\mathcal{O}(N^2)$ that is prohibitive for practical applications. Definition of the tree determines the absence of loop among the connections of vertices. According to this property, a well-designed dynamic programming algorithm can be used to speed up the optimization and inference process.

We introduce two sequential passes, namely ***aggregation*** and ***propagation***, which are performed by traversing the tree structure. Let one vertex to be the root node. In the aggregation pass, the process is traced from the leaf nodes to the root node in the tree. For a vertex, its features do not update until all the children have been visited. In the propagation pass, the features will propagate from the updated

vertex to their children recursively.

$$\text{Aggr}(\boldsymbol{\xi})_i = \boldsymbol{\xi}_i + \sum_{\text{par}(j)=i} S\left(\boldsymbol{E}_{i,j}\right) \text{Aggr}(\boldsymbol{\xi})_j. \tag{6}$$

$$\text{Prop}(\boldsymbol{\xi})_i = \begin{cases} \text{Aggr}(\boldsymbol{\xi})_r & i = r \\ S\left(\boldsymbol{E}_{\text{par}(i),i}\right) \text{Prop}(\boldsymbol{\xi})_{\text{par}(i)} + \left(1 - S^2\left(\boldsymbol{E}_{i,\text{par}(i)}\right)\right) \text{Aggr}(\boldsymbol{\xi})_i & i \neq r \end{cases} \tag{7}$$

The sequential passes can be formulated into the recursive operators for the input $\boldsymbol{\xi}$: the *aggregation pass* and the *propagation pass* are respectively illustrated in Eq. 6 and Eq. 7, where par($i$) indicates the parent of vertex $i$ in the tree whose root is vertex $r$. $\text{Prop}(\boldsymbol{\xi})_r$ is initialized from the updated value $\text{Aggr}(\boldsymbol{\xi})_r$ of root vertex $r$.

---

**Algorithm 1** Linear time algorithm for Learnable Tree Filter

---

**Input:** Tree $G \in \mathbb{N}^{(N-1)\times 2}$; Input feature $\boldsymbol{x} \in \mathbb{R}^{C\times N}$; Pairwise distance $\boldsymbol{\omega} \in \mathbb{R}^N$; Gradient of loss w.r.t. output feature $\boldsymbol{\phi} \in \mathbb{R}^{C\times N}$; channel $C$, vertex $N$; Set of vertices $\Omega$.

**Output:** Output feature $\boldsymbol{y}$; Gradient of loss w.r.t. input feature $\frac{\partial loss}{\partial \boldsymbol{x}}$, w.r.t. pairwise distance $\frac{\partial loss}{\partial \boldsymbol{\omega}}$.

**Preparation:**

$r = \text{Uniform}(\Omega)$             ▷ *Root vertex sampled with uniform distribution*

$\boldsymbol{T} = \text{BFS}(G, r)$             ▷ *Breadth-first topological order for Aggr and Prop*

$\boldsymbol{J} = \mathbf{1} \in \mathbb{R}^{1\times N}$             ▷ *All-ones matrix for normalization coefficient*

**Forward:**

1. $\{\hat{\boldsymbol{\rho}}, \hat{z}\} = \text{Aggr}(\{f(\boldsymbol{x}), \boldsymbol{J}\})$           ▷ *Aggregation from leaves to root*

2. $\{\boldsymbol{\rho}, z\} = \text{Prop}(\{\hat{\boldsymbol{\rho}}, \hat{z}\})$           ▷ *Propagation from root to leaves*

3. $\boldsymbol{y} = \boldsymbol{\rho}/z$           ▷ *Normalized output feature*

**Backward:**

1. $\{\hat{\boldsymbol{\psi}}, \hat{\boldsymbol{\nu}}\} = \text{Aggr}(\{\boldsymbol{\phi}/z, \boldsymbol{\phi}\cdot\boldsymbol{y}/z\})$           ▷ *Aggregation from leaves to root*

2. $\{\boldsymbol{\psi}, \boldsymbol{\nu}\} = \text{Prop}(\{\hat{\boldsymbol{\psi}}, \hat{\boldsymbol{\nu}}\})$           ▷ *Propagation from root to leaves*

3. $\frac{\partial loss}{\partial \boldsymbol{x}} = \frac{\partial f(\boldsymbol{x})}{\boldsymbol{x}}\cdot\boldsymbol{\psi}$           ▷ *Gradient of loss w.r.t. input feature*

4. **for** $i \in \boldsymbol{T}\backslash r$ **do**

     $j = \text{par}(i)$           ▷ *Parent of vertex $i$*

     $\boldsymbol{\gamma}_i^s = \hat{\boldsymbol{\psi}}_i\cdot\boldsymbol{\rho}_i + \boldsymbol{\psi}_i\cdot\hat{\boldsymbol{\rho}}_i - 2S(\boldsymbol{E}_{i,j})\hat{\boldsymbol{\psi}}_i\cdot\hat{\boldsymbol{\rho}}_i$    ▷ *Gradient of unnormalized output feature*

     $\boldsymbol{\gamma}_i^z = \hat{\boldsymbol{\nu}}_i z_i + \boldsymbol{\nu}_i \hat{z}_i - 2S(\boldsymbol{E}_{i,j})\hat{\boldsymbol{\nu}}_i\hat{z}_i$        ▷ *Gradient of normalization coefficient*

     $\frac{\partial loss}{\partial \boldsymbol{\omega}_{i,j}} = \frac{\partial S(\boldsymbol{E}_{i,j})}{\partial \boldsymbol{\omega}_{i,j}}\sum\left(\boldsymbol{\gamma}_i^s - \boldsymbol{\gamma}_i^z\right)$      ▷ *Gradient of loss w.r.t. pairwise distance*

    **end**

---

As shown in the algorithm 1, we propose a *linear-time* algorithm for the tree filtering module, whose proofs are provided in the appendix. In the preparation stage, we uniformly sample a vertex as the root and perform breadth-first sorting (BFS) algorithm to obtain the topological order of tree $G$. The BFS algorithm can be accelerated by the parallel version on GPUs and ensure the efficiency of the following operations.

To compute the normalization coefficient, we construct an all-ones matrix as $\boldsymbol{J}$. Since the embedded feature $f(\boldsymbol{x})$ is independent of the matrix $\boldsymbol{J}$, the forward computation can be factorized into two dynamic programming processes. Furthermore, we propose an efficient implementation for the backward process. To reduce the unnecessary intermediate process, we combine the gradient of module and loss function. Note that the output $\boldsymbol{y}$ and normalization coefficient $z$ have been already computed in the inference phase. Thus the key of the backward process is to compute the intermediate variables $\boldsymbol{\psi}$ and $\boldsymbol{\nu}$. The computation of these variables can be accelerated by the proposed linear time algorithm. And we adopt another iteration process for the gradient of pairwise distance.

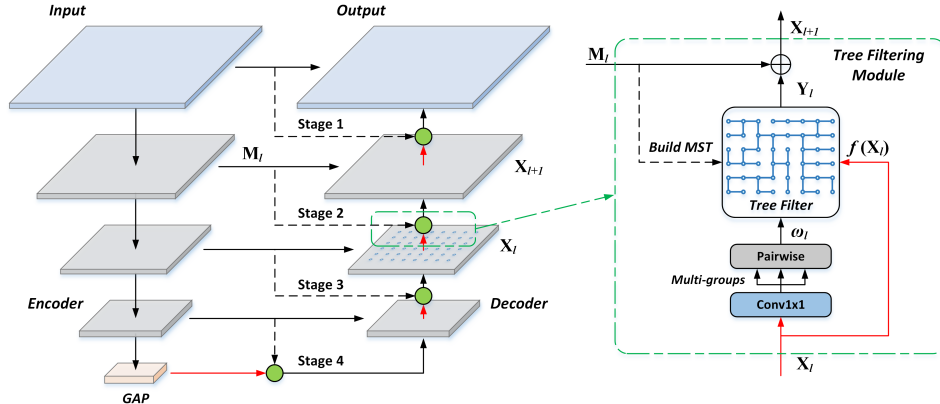

Figure 2: Overview of the proposed framework for semantic segmentation. The network is composed of a backbone encoder and a naive decoder. **GAP** denotes the extra global average pooling block. **Multi-groups** means using different feature splits to generate multiple groups of tree weights. The right diagram elaborates on the process details of *a single stage tree filtering module*, denoted as the green node in the decoder. Red arrows represent upsample operations. Best viewed in color.

**Computational complexity.** Since the number of batch and channel is much smaller than the vertices in the input feature, we only consider the influence of the vertices. For each channel, the computational complexity of all the processes in algorithm 1, including the construction process of MST and the computation of pairwise distance, is $\mathcal{O}(N)$, which is linearly dependent on the number of vertices. It is necessary to point out that MST can be built in *linear* time using *Contractive Borůvka algorithm* if given a *planar* graph, as is designed in this paper. Note that the batches and channels are independent of each other. For the practical implementation on GPUs, we can naturally perform the algorithm for batches and channels in parallel. Also, we adopt an effective scheduling scheme to compute vertices of the same depth on the tree parallelly. Consequently, the proposed algorithm reduces the computational complexity and time consumption dramatically.

## 2.3 Network Architecture for Segmentation

Based on the efficient implementation algorithm, the proposed tree filtering module can be easily embedded into deep neural networks for resource-friendly feature aggregation. To illustrate the effectiveness of the proposed module, here we employ ResNet [8] as our encoder to build up a unified network. The encoded features from ResNet are usually computed with output stride 32. To remedy for the resolution damage, we design a naive decoder module following previous works [14, 15]. In details, the features in the decoder are gradually upsampled by a factor of 2 and summed with corresponding low-level features in the encoder, similar to that in FPN [25]. After that, the bottom-up embedding functions in the decoder are replaced by tree filter modules for multi-scale feature transform, as intuitively illustrated in Fig. 2.

To be more specific, given a low-level feature map $M_l$ in top-down pathway, which riches in instance details, a 4-connected planar graph can be constructed easily with the guidance of $M_l$. Then, the edges with substantial dissimilarity are removed to formulate MST using the *Borůvka* [26] algorithm. High level semantic cues contained in $X_l$ are extracted using a simplified embedding function (Conv $1 \times 1$). To measure the *pairwise dissimilarity* $\omega$ in Eq. 2 ($\omega_l$ in Fig. 2), the widely used Euclidean distance [27] is adopted. Furthermore, different groups of tree weights $\omega_l$ are generated to capture component dependent features, which will be analyzed in Sec 3.2. To highlight the effectiveness of the proposed method, the feature transformation $f(\cdot)$ is simplified to identity mapping, where $f(X_l) = X_l$. Thus, the learnable tree filter can be formulated by the algorithm elaborated on Sec. 2.1. Finally, the low-level feature map $M_l$ is fused with the operation output $y_l$ using pixel-wise summation. For multi-stage feature aggregation, the building blocks (green nodes in Fig. 1) are applied to different resolutions (Stage 1 to 3 in Fig. 1). An extra global average pooling operation is added to capture global context and construct another tree filtering module (Stage 4). The promotion brought by the extra components will be detailed discussed in ablation studies.

# 3 Experiments

In this section, we firstly describe the implementation details. Then the proposed approach will be decomposed step-by-step to reveal the effect of each component. Comparisons with several state-of-the-art benchmarks on PASCAL VOC 2012 [23] and Cityscapes [24] are presented at last.

## 3.1 Implementation Details

Following traditional protocols [3, 15, 28], ResNet [8] is adopted as our backbone for the following experiments. Specifically, we employ the "poly" schedule with an initial learning rate 0.004 and power 0.9. The networks are optimized for 40K iterations using mini-batch stochastic gradient descent (SGD) with a weight decay of 1e-4 and a momentum of 0.9. We construct each mini-batch for training from 32 random crops ($512 \times 512$ for PASCAL VOC 2012 [23] and $800 \times 800$ for Cityscapes [24]) after randomly flipping and scaling each image by 0.5 to $2.0\times$.

## 3.2 Ablation Studies

To elaborate on the effectiveness of the proposed approach, we conduct extensive ablation studies. First, we give detailed structure-preserving relation analysis as well as the visualization, as presented in Fig. 3. Next, the equipped stage and group number of the tree filtering module is explored. Different building blocks are compared to illustrate the effectiveness and efficiency of the tree filtering module.

**Structure-preserving relations.** As intuitively presented in the first row of Fig. 3, given different positions, the corresponding instance details are fully activated with the high response, which means that the proposed module has learned object structures and the long-range intra-class dependencies. Specifically, the object details (*e.g.*, boundaries of the *train* rather than the coarse regions in Non-local [13] blocks) have been highlighted in the affinity maps. Qualitative results are also given to illustrate the preserved structural details, as presented in Fig. 4.

**Which stage to equip the module?** Tab. 1 presents the results when applying the tree filtering module to different stages (group number is fixed to 1). The convolutional operations are replaced by the building block (green nodes in Fig. 2) to form the *equipped stage*. As can be seen from Tab. 1, the

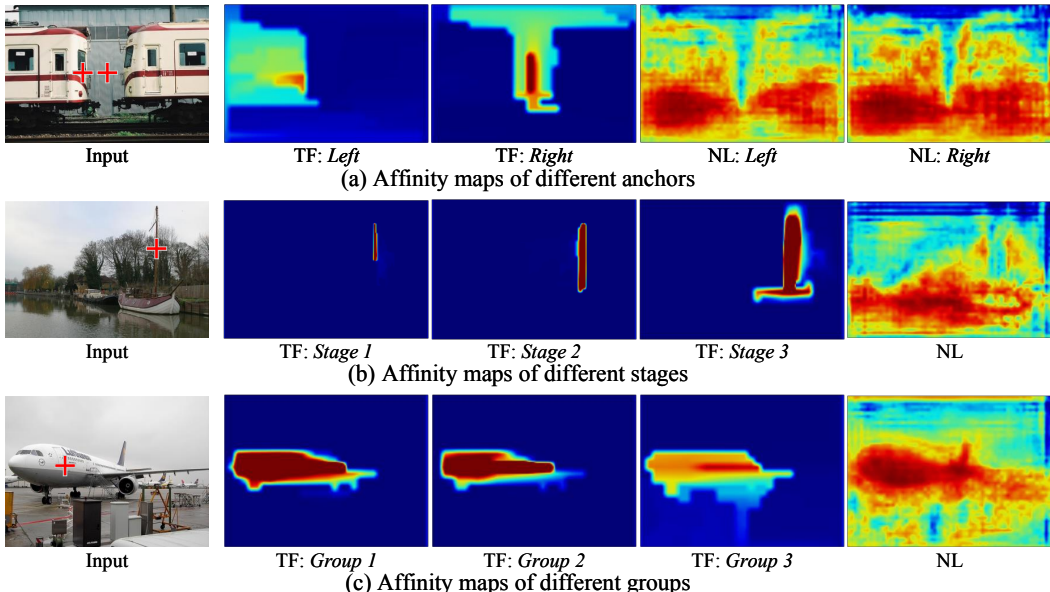

Figure 3: Visualization of affinity maps in the specific position (marked by the red cross in each input image). **TF** and **NL** denotes using the proposed *Tree Filtering module* and *Non-local block* [13], respectively. Different positions, resolution stages, and selected groups are explored in (a), (b), and (c), respectively. Our approach preserves more detailed structures than Non-local block. All of the input images are sampled from PASCAL VOC 2012 *val* set.

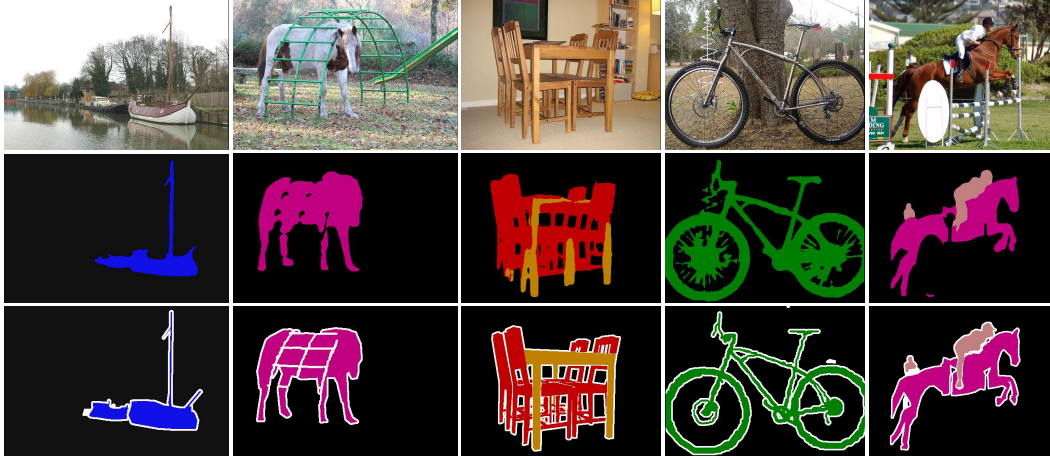

Figure 4: Qualitative results on PASCAL VOC 2012 *val* set. Given an input image from the top row, the structural cues are preserved in the corresponding prediction (the middle row). The generated results contains rich details even compared with its ground truth in the bottom row.

network performance consistently improves with more tree filtering modules equipped. This can also be concluded from the qualitative results (the second row in Fig. 3), where the higher stage (Stage 3) contains more semantic cues and lower stages (Stage 1 and 2) focus more on complementary details. The maximum gap between the multi-stage equipped network and the raw one even up to **4.5%** based on ResNet-50 backbone. Even with the powerful ResNet-101, the proposed approach still attains a 2.1% absolute gain, reaching 77.1% mIoU on PASCAL VOC 2012 *val* set.

**Different group numbers.** Different group settings are used to generate weights for the single-stage tree filtering module when fixing the channel number to 256. As shown in Tab. 2, the network reaches top performance (Group Num=16) when the group number approaching the category number (21 in PASCAL VOC 2012), and additional groups afford no more contribution. We guess the reason is that different kinds of tree weights are needed to deal with similar but different components, as shown in the third row of Fig. 3.

**Different building blocks.** We further compare the proposed tree filtering module with classic context modeling blocks (*e.g.,* PSP block [3] and Non-local block [13]) and prove the superiority both in *accuracy* and *efficiency*. As illustrated in Tab. 3, the proposed module (TF) achieves better performance than others (PSP and NL) with much less resource consumption. What's more, the tree filtering module brings consistent improvements in different backbones (**5.2%** for ResNet-50 with stride 32 and 2.6% for ResNet-101) with additional **0.7M** parameters and **1.3G** FLOPs overheads. Due to the structural-preserving property, the proposed module achieves additional 1.1% improvement over the PSP block with neglectable consumption, as presented in Tab. 3. And the extra Non-local block contributes no more gain over the tree filtering module, which could be attributed to the already modeled feature dependencies in the tree filter.

**Extra components.** Same with other works [15, 28], we adopt some simple components for further improvements, including an extra *global average pooling* operation and additional *ResBlocks* in the decoder. In details, the global average pooling block combined with the Stage 4 module (see Fig. 2) is added after the backbone to capture global context, and the "Conv1×1" operations in the decoder (refer to the detailed diagram in Fig. 2) are replaced by ResBlocks (with "Conv3x3"). As presented in Tab. 4, the proposed method achieves consistent improvements and attains **79.4%** on PASCAL VOC 2012 *val* set without applying data augmentation strategies.

### 3.3 Experiments on Cityscapes

To further illustrate the effectiveness of the proposed method, we evaluate the Cityscapes [24] dataset. Our experiments involve the 2975, 500, 1525 images in *train*, *val*, and *test* set, respectively. With multi-scale and flipping strategy, the proposed method achieves **80.8%** mIoU on Cityscapes *test* set

Table 1: Comparisons among different stages to equip tree filtering module on PASCAL VOC 2012 *val* set when using the proposed decoder architecture. Multi-scale and flipping strategy are adopted for testing.

| Backbone | Stage | mIoU (%) |
|---|---|---|
| ResNet-50 | None | 70.2 |
| | + Stage 1 | 72.1 |
| | + Stage 1-2 | 73.1 |
| | + Stage 1-3 | 74.7 |
| ResNet-101 | None | 75.0 |
| | + Stage 1-3 | **77.1** |

Table 2: Comparisons among different group settings of tree filtering module on PASCAL VOC 2012 *val* set when using the proposed decoder structure. Multi-scale and flipping strategy are adopted for testing.

| Backbone | Group Num | mIoU (%) |
|---|---|---|
| ResNet-50 | 0 | 70.2 |
| | 1 | 72.1 |
| | 4 | 73.2 |
| | 8 | 74.0 |
| | 16 | **74.4** |
| | 32 | 74.4 |

Table 3: Comparisons among different building blocks on PASCAL VOC 2012 *val* set when using ResNet-50 as feature extractor *without* decoder. **TF**, **PSP**, and **NL** denotes using the proposed *Tree Filtering module*, *PSP block* [3], and *Non-local block* [13] as the building block, respectively. **OS** represents the output stride used in the backbone. We calculate FLOPs when given a single scale $512 \times 512$ input. All of the data augmentation strategies are dropped.

| Backbone | Block | Decoder | OS | Params (M) | FLOPs (G) | Δ FLOPs (G) | mIoU (%) |
|---|---|---|---|---|---|---|---|
| ResNet-50 | None | ✗ | 8 | 129.3 | 162.1 | 0.0 | 69.2 |
| | NL | ✗ | 8 | 158.3 | 199.1 | +37.0 | 74.2 |
| | PSP | ✗ | 8 | 178.3 | 171.1 | +9.0 | 74.3 |
| | TF | ✗ | 8 | 133.3 | 163.1 | +1.0 | 74.9 |
| | NL+TF | ✗ | 8 | 162.3 | 200.1 | +38.0 | 74.9 |
| | PSP+TF | ✗ | 8 | 182.3 | 172.1 | +10.0 | 75.4 |
| ResNet-50 | None | ✓ | 32 | 102.0 | 39.2 | 0.0 | 67.3 |
| | TF | ✓ | 32 | 102.7 | 40.5 | +1.3 | 72.5 |
| ResNet-101 | None | ✓ | 32 | 175.0 | 65.0 | 0.0 | 72.8 |
| | **TF** | ✓ | 32 | 175.7 | 66.3 | +1.3 | **75.4** |

Table 4: Comparisons among different components on PASCAL VOC 2012 *val* set. **TF(multi)** denotes *multi-stage Tree Filtering modules* with decoder. **Extra** represents extra components. All of the data augmentation strategies are dropped.

| Backbone | TF(multi) | Extra | mIoU (%) |
|---|---|---|---|
| ResNet-50 | ✗ | ✗ | 67.3 |
| | ✗ | ✓ | 72.5 |
| | ✓ | ✗ | 72.5 |
| | ✓ | ✓ | 75.6 |
| ResNet-101 | ✗ | ✓ | 78.3 |
| | ✓ | ✓ | 79.4 |

Table 5: Comparisons with state-of-the-arts results on Cityscapes *test* set trained on *fine* annotation. We adopt *vanilla* ResNet-101 as our backbone.

| Method | Backbone | mIoU (%) |
|---|---|---|
| RefineNet [29] | ResNet-101 | 73.6 |
| DSSPN [30] | ResNet-101 | 77.8 |
| PSPNet [3] | ResNet-101 | 78.4 |
| BiSeNet [31] | ResNet-101 | 78.9 |
| DFN [28] | ResNet-101 | 79.3 |
| PSANet [32] | ResNet-101 | 80.1 |
| DenseASPP [33] | DenseNet-161 | 80.6 |
| **Ours** | ResNet-101 | **80.8** |

when trained on *fine* annotation data only. Compared with previous leading algorithms, our method achieves superior performance using ResNet-101 without bells-and-whistles.

## 3.4 Experiments on PASCAL VOC

We carry out experiments on PASCAL VOC 2012 [23] that contains 20 object categories and one background class. Following the procedure of [28], we use the augmented data [38] with 10,582 images for training and raw *train-val* set for further fine-tuning. In inference stage, multi-scale and horizontally flipping strategy are adopted for data augmentation. As shown in Tab. 6, the proposed method achieves the state-of-the-art performance on PASCAL VOC 2012 [23] *test* set. In details, our

Table 6: Comparisons with state-of-the-arts results on PASCAL VOC 2012 *test* set. We adopt *vanilla* ResNet-101 without atrous convolutions as our backbone.

| without MS-COCO pretrain | | | with MS-COCO pretrain | | |
|---|---|---|---|---|---|
| Method | Backbone | mIoU (%) | Method | Backbone | mIoU (%) |
| FCN [1] | VGG 16 | 62.2 | GCN [36] | ResNet-152 | 82.2 |
| Deeplab v2 [2] | VGG 16 | 71.6 | RefineNet [29] | ResNet-101 | 84.2 |
| DPN [34] | VGG 16 | 74.1 | PSPNet [3] | ResNet-101 | 85.4 |
| Piecewise [35] | VGG 16 | 75.3 | Deeplab v3 [14] | ResNet-101 | 85.7 |
| PSPNet [3] | ResNet-101 | 82.6 | EncNet [4] | ResNet-101 | 85.9 |
| DFN [28] | ResNet-101 | 82.7 | DFN [28] | ResNet-101 | 86.2 |
| EncNet [4] | ResNet-101 | 82.9 | ExFuse [37] | ResNet-101 | 86.2 |
| **Ours** | ResNet-101 | **84.2** | **Ours** | ResNet-101 | **86.3** |

approach reaches **84.2%** mIoU without MS-COCO [39] pre-train when adopting *vanilla* ResNet-101 as our backbone. If MS-COCO [39] is added for pre-training, our approach achieves the leading performance with **86.3%** mIoU.

## 4 Conclusion

In this work, we propose the learnable tree filter for structure-preserving feature transform. Different from most existing methods, the proposed approach leverages *tree-structured* graph for long-range dependencies modeling while preserving detailed object structures. We formulate the tree filtering module and give an efficient implementation with linear-time source consumption. Extensive ablation studies have been conducted to elaborate on the effectiveness and efficiency of the proposed method, which is proved to bring consistent improvements on different backbones with little computational overhead. Experiments on PASCAL VOC 2012 and Cityscapes prove the superiority of the proposed approach on semantic segmentation. More potential domains with structure relations (*e.g.*, detection and instance segmentation) remain to be explored in the future.

## 5 Acknowledgment

We would like to thank Lingxi Xie for his valuable suggestions. This research was supported by the National Key R&D Program of China (No. 2017YFA0700800).

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
