[Supplementary Material]

# A Algorithm Proof

In this section, we present the detailed proofs for the formula in Algorithm 1. Note that the symbols follow the definition in the main paper.

**Proof 1** *Given one vertex as $i$, the Eq. 8 proves the backward process of $\frac{\partial loss}{\partial \boldsymbol{x}}$ in Algorithm 1.*

$$
\begin{aligned}
\frac{\partial loss}{\partial \boldsymbol{x}_i} &= \frac{\partial f(\boldsymbol{x}_i)}{\partial \boldsymbol{x}_i} \sum_{j \in \Omega} \frac{\partial loss}{\partial \boldsymbol{y}_j} \frac{\partial \boldsymbol{y}_j}{\partial \boldsymbol{x}_i} \\
&= \frac{\partial f(\boldsymbol{x}_i)}{\partial \boldsymbol{x}_i} \sum_{j \in \Omega} S(\boldsymbol{E}_{i,j})(\frac{\partial loss}{\partial \boldsymbol{y}_j} \frac{1}{z_j}) \\
&= \frac{\partial f(\boldsymbol{x}_i)}{\partial \boldsymbol{x}_i} \boldsymbol{\psi}_i
\end{aligned}
\tag{8}
$$

**Proof 2** *Given one edge with a pair of connected vertices $i$ and $j$, the Eq. 9-11 proves the backward process of $\frac{\partial loss}{\partial \boldsymbol{\omega}}$ in Algorithm 1, where $\Omega_i$ indicates the set of children of vertex $i$ in the tree whose root is $j$ and $\Omega_j$ indicates the set of children of vertex $j$ in the tree whose root is $i$.*

$$
\begin{aligned}
\frac{\partial loss}{\partial \boldsymbol{\omega}_{i,j}} &= \frac{\partial S(\boldsymbol{E}_{i,j})}{\partial \boldsymbol{\omega}_{i,j}} \sum \sum_{m \in \Omega} \frac{\partial loss}{\partial \boldsymbol{y}_m} \frac{\partial \boldsymbol{y}_m}{\partial S(\boldsymbol{E}_{i,j})} \\
&= \frac{\partial S(\boldsymbol{E}_{i,j})}{\partial \boldsymbol{\omega}_{i,j}} \sum (\sum_{m \in \Omega} \frac{1}{z_m} \frac{\partial loss}{\partial \boldsymbol{y}_m} \frac{\partial z_m \boldsymbol{y}_m}{\partial S(\boldsymbol{E}_{i,j})} - \sum_{m \in \Omega} \frac{\partial loss}{\partial \boldsymbol{y}_m} \frac{\boldsymbol{y}_m}{z_m} \frac{\partial z_m}{\partial S(\boldsymbol{E}_{i,j})}) \\
&= \frac{\partial S(\boldsymbol{E}_{i,j})}{\partial \boldsymbol{\omega}_{i,j}} \sum (\boldsymbol{\gamma}_i^s - \boldsymbol{\gamma}_i^z)
\end{aligned}
\tag{9}
$$

*The computational complexity of the component $\boldsymbol{\gamma}_i^s$ can be reduce to linear by the dynamic programming procedure below.*

$$
\begin{aligned}
\boldsymbol{\gamma}_i^s &= \sum_{m \in \Omega} \frac{\partial loss}{\partial \boldsymbol{y}_m} \frac{1}{z_m} \frac{\partial z_m \boldsymbol{y}_m}{\partial S(\boldsymbol{E}_{i,j})} \\
&= \sum_{m \in \Omega} (\frac{\boldsymbol{\phi}}{z})_m \sum_{k \in \Omega} \frac{\partial S(\boldsymbol{E}_{m,k}) f(\boldsymbol{x}_k)}{\partial S(\boldsymbol{E}_{i,j})} \\
&= \sum_{m \in \Omega_i} (\frac{\boldsymbol{\phi}}{z})_m \sum_{k \in \Omega_j} S(\boldsymbol{E}_{m,i}) S(\boldsymbol{E}_{i,k}) f(\boldsymbol{x}_k) + \sum_{m \in \Omega_j} (\frac{\boldsymbol{\phi}}{z})_m \sum_{k \in \Omega_i} S(\boldsymbol{E}_{m,j}) S(\boldsymbol{E}_{j,k}) f(\boldsymbol{x}_k) \\
&= \sum_{m \in \Omega_i} S(\boldsymbol{E}_{m,i})(\frac{\boldsymbol{\phi}}{z})_m \sum_{k \in \Omega_j} S(\boldsymbol{E}_{i,k}) f(\boldsymbol{x}_k) + \sum_{m \in \Omega_j} S(\boldsymbol{E}_{m,j})(\frac{\boldsymbol{\phi}}{z})_m \sum_{k \in \Omega_i} S(\boldsymbol{E}_{j,k}) f(\boldsymbol{x}_k) \\
&= \hat{\boldsymbol{\psi}}_i \cdot \hat{\boldsymbol{\rho}}_j + \hat{\boldsymbol{\psi}}_j \cdot \hat{\boldsymbol{\rho}}_i \\
&= \hat{\boldsymbol{\psi}}_i \cdot (\boldsymbol{\rho}_i - S(\boldsymbol{E}_{i,j}) \hat{\boldsymbol{\rho}}_i)) + \hat{\boldsymbol{\rho}}_i \cdot (\boldsymbol{\psi}_i - S(\boldsymbol{E}_{j,i}) \hat{\boldsymbol{\psi}}_i)) \\
&= \hat{\boldsymbol{\psi}}_i \cdot \boldsymbol{\rho}_i + \boldsymbol{\psi}_i \cdot \hat{\boldsymbol{\rho}}_i - 2S(\mathbf{E}_{i,j}) \hat{\boldsymbol{\psi}}_i \cdot \hat{\boldsymbol{\rho}}_i
\end{aligned}
\tag{10}
$$

*The same procedure can be easily adapted to obtain the component $\boldsymbol{\gamma}_i^z$.*

$$
\boldsymbol{\gamma}_i^z = \sum_{m \in \Omega} \frac{\partial loss}{\partial \boldsymbol{y}_m} \frac{\boldsymbol{y}_m}{z_m} \frac{\partial z_m}{\partial S(\boldsymbol{E}_{i,j})} = \hat{\boldsymbol{\nu}}_i z_i + \boldsymbol{\nu}_i \hat{z}_i - 2S(\boldsymbol{E}_{i,j}) \hat{\boldsymbol{\nu}}_i \hat{z}_i
\tag{11}
$$