[Reviews · NeurIPS 2019]

Reviewer 1



1. Originality: The idea is motivated by Yang, stereo matching using tree filtering, PAMI2015. But it is considerably novel in incorporating this idea in an end-to-end deep network. 2. Quality: + Tree structure to exploit irregular shaped long term dependencies. + Detailed ablation analysis. + More efficient than PSP and non-local operations. 3. Clarity: This paper is in general well written. But I still have some concerns to be clarified: - How the higher level semantics/features can guide to generate the MST (L40-41, L141-143)? In my understanding, is the MST generate based on the weights calculated by the CURRENT feature map rather than higher ones? - How to initialize the MST weights? Are they initialized once (by Euclidean distance on features of each pixel) then updated solely by gradients, or the Euclidean distance should be calculated every step (as the CNN paras change)? - What is multi-groups in Figure 2 and how does it take efforts? 4. Significance: The proposed method better exploits irregular shaped long term dependencies for semantic segmentation. ---------------- I appreciate the authors' rebuttal which well addressed my concerns, therefore, I remain my recommendation as an accept.

Reviewer 2



Novelty ====== This paper introduces a clear new module into the crowded space of contextual models for semantic segmentation. In addition to the value of the core idea, the authors explain how to differentiate their tree filtering operator, enabling end-to-end training. Finally, they also propose a technically interesting way to reduce the complexity of computing their tree filtering module down to linear in the number of pixels. Experiments ========== The experiments are thorough, with lots of comparisons to the state-of-the-art and a clear ablation study. However, the improvements over the state-of-the-art are very minor: +0.6% mIoU over PSP [12] on VOC12 val (table 3), +0.2% mIoU over DenseASPP [32] on Cityscapes test (table 5), and +0.1% mIoU over ExFuse [36] on VOC12 test with MS-COCO pretrain (table 6). In the light of these numbers, I find the results oversold: the abstract states "leading performance on VOC 2012 (86.3% mIoU) and Cityscapes (80.8% mIoU)", exactly referring to the last two results I listed in this review (+0.1% and +0.2%). Moreover, the conclusion states "superiority of the proposed method on VOC12 and Cityscapes". These claims are not justified by such a small delta. It would be small for any system, but especially for neural networks, due to the inherent randomness of their SGD training procedure. The one significant result I could find is +1.3% on VOC12 test set without MS-COCO pretraining, but that's not what is sold in the abstract. The ablation studies in Tables 1 and 2 show the impact of the proposed new module starting from a system with no context at all. The effect is nice (+2.1% on ResNet101 on VOC12), but the true comparison is: how much more does this new context module bring compared to previous ones? And the cleanest answer I could find is in Table 3: +0.6% over PSP [3] and +0.7% over NL [12]. That is a small effect. Quality of writing ============= The quality of writing is good, but not great. There are many missing or oddly places articles, incorrect verb conjugations, and so on. Summary ======= This paper offers good novelty and it is technically interesting. However, the results are underwhelming and the comparisons to the state-of-the-art are very oversold. Reaction to rebuttal =============== The authors have provided some reply to my critique that the improvements over the state-of-the-art are small, mainly with one new experiment and by promising to include dilated convolutions to improve performance of their backbone network. Most importantly, they promised to revise their claims in the final version. This is very important. If the delta improvements remain as small as they were in the submission version, then their claims must be toned down. In the light of the rebuttal, I keep my original score and trust the authors to keep their promise.

Reviewer 3



Originality: The proposed approach is motivated by the traditional tree filter [reference 18 in the paper]. While the idea of combining a combinatorial optimization algorithm with (deep) learning has become quite common, the particular approach poses challenges implementationwise and provides many benefits, as it allows to model long-range spatial dependencies in an efficient way, a very desirable property. Quality: The paper contains sufficient ablation studies and experiments that demonstrate the strengths of the method in terms of its differences to other recent methods. One critique is a claim about the computational complexity of the proposed algorithm made in lines 124-125: "...the computational complexity of all the processes in algorithm 1, including the construction process of minimum spanning tree and the computation of pairwise distance, is O(N)...". It is mentioned in line 143 that the Boruvka algorithm is used (reference 25 in the paper). Nevertheless, the Boruvka algorithm is O(N log N). The authors should either clarify or correct this claim. While there is a randomized algorithm that achieves linear complexity in expectation, this is not mentioned in the paper. Clarity: The paper is clearly written and well organized. There are a few cases of language misuse and typos, but these can be easily fixed. Some examples: Line 1: "plays a vital role in semantic segmentation. Most of the existing methods..." Line 116: "Also, we propose..." Line 120: "The computation of these variables can be accelerated..." Significance: The idea of exploiting structure in images within a learning framework (as a neural network) has been desired for a long time. It is known that convolutional neural networks can increase the receptive field with depth but the effective receptive field remains concentrated. In recent years, various graph neural networks have been proposed and studied extensively, but an efficient / linear time algorithm was missing. Non-local networks have also been proposed but due to their O(N^2) complexity, they are only applicable for small graphs. I believe the proposed approach is very likely to be used and build upon by other researchers.

[Author Response · NeurIPS 2019]

# Author Rebuttal for NeurIPS 2019 Submission #964

We thank all the reviewers for their positive comments and valuable suggestions. This paper presents a linear-time *learnable tree filter* to capture long-range dependencies while preserving structural details for semantic segmentation. In the initial review, all reviewers agreed with the novelty and contributions of the *learnable tree filter*. We respond to all the comments as follows. In addition, we will carefully revise the manuscript to improve its readability before the final submission. As required by R5, an executable example (the same anonymous link as that in Supplementary Materials) has been provided. **All the source code will be released to the community soon.**

## Response to Reviewer #1

**Q1:** *How to generate MST and initialize the weights? Details of muti-groups in Fig. 2?*

**A1:** Sorry for the confusion. Actually, the weights used for *MST construction* and *filtering* are different. The MST is constructed upon the low-level feature in encoder (defined as $M_l$ in L141 and illustrated in Fig. 2). The weights for filtering do not need to be initialized, but are calculated based on high-level semantics (defined as $X_l$ in L143 and illustrated in Fig. 2) in *every* step. *Multi-groups* in Fig. 2 represents calculating multiple groups of weights for the filter, which allows it to be sensitive to different components (please refer to L145-146 and Sec. 3.2 for more details).

**Q2:** *Can the proposed tree filter be used in encoder? How would the results be like?*

**A2:** Yes, it can be used in encoder. Actually, the experiments in Tab. 3 (when OS is 8) are conducted with encoder *only*. In addition, we inject the *Tree Filtering module* into C3-C5 of ResNet-50 and achieve a 71.7% mIoU, which is *inferior* to that in Tab. 3 (a 72.5% mIoU when applied to the decoder). It means higher-level semantic matters for the filter weights (decoder contains richer semantic cues, identical with the design in L40-43). This will be made clear.

**Q3:** *The evaluations on ADE20K?*

**A3:** Thanks for this suggestion. During rebuttal, we evaluate our methods using a ResNet-50 backbone on the ADE20K *val* set, and list the results in Tab. 7. This table will be added in the final version.

Figure 5: Runtime comparisons on Tesla V100.

| Backbone | TF | MS | mIoU (%) | Pixel Acc (%) |
|----------|----|----|----------|---------------|
|  | ✗ | ✗ | 35.0 | 76.5 |
| ResNet-50 | ✓ | ✗ | 40.0 | 79.3 |
|  | ✓ | ✓ | **41.1** | **80.2** |

Table 7: Results on the ADE20K *val* set. **TF** denotes *multi-stage Tree Filtering modules* with decoder. **MS** indicates multi-scale testing strategy. All the experiments are conducted with a *vanilla* ResNet-50 backbone (*without* dilated convolutions or the ASPP module).

## Response to Reviewer #4

**Q1:** *Additional improvements over the PSP and Non-local module?*

**A1:** Different from PSP [3] and Non-local [12] module, the proposed *Tree Filtering module* also preserves structural details when capturing long-range dependencies (refer to the qualitative analysis in Ablation Study and Supplementary Materials). To verify the effectiveness, we conduct experiments with additional PSP and Non-local module, and achieve 1.1% and 0.7% absolute gain (**75.4%** for PSP+TF and **74.9%** for NL+TF), respectively.

**Q2:** *Performance comparison.*

**A2:** Thanks for your comments. Actually, we adopt *vanilla* ResNet-101 *without* dilated convolutions or Dense blocks (adopted by PSANet [31] and DenseASPP [32]) as our backbone in Tab. 5 and Tab. 6. Of course, we will give more competitive results with stronger backbones as well as revise our claim in the final version.

## Response to Reviewer #5

**Q1:** *Why the computational complexity of MST construction (Borůvka's algorithm) is linear?*

**A1:** Sorry for not having clarified this point clearly. Generally, the *Borůvka's algorithm* runs in $\mathcal{O}(E \log V)$. Nevertheless, as illustrated in Fig. 1, we build MST from a *4-connected **planar** graph*. When the input graph is planar, the computational complexity can be reduced to *linear* using *Contractive Borůvka's algorithm*. Of course, this can also be achieved by other algorithms (*e.g.,* [Karger *et.al.*, JACM, 1995]). We will clarify it in the final version.

**Q2:** *Provide the minimal working example and compare empirical runtime with other methods.*

**A2:** Following your suggestion, we have already provided the **executable code** (the same anonymous link with that in Supplementary Materials) as well as a benchmark for runtime comparison. We also illustrate the comparison against the non-local operation in Fig. 5. To clarify more details, we will release our source code to the community.

**Q3:** *How to select hyper-parameters?*

**A3:** For network training, we just follow traditional protocols (refer to Sec. 3.1) without bells-and-whistles. While for the proposed *Tree Filtering module*, we conduct ablation studies (especially for the *equipped stage* and *group number* in Sec. 3.2) and choose the best-performed combination of hyper-parameters.

[Meta-Review · NeurIPS 2019]

The paper introduces learnable tree filter using minimal spanning tree for modeling long-range dependencies. The proposed algorithm is linear-time and can be incorporated into commonly used deep neural network. Empirical evaluation shows leading performance with ResNet-101 on PASCAL VOC 2012. All reviewers found the contributions of this work significant, both from methodological and empirical perspectives, and rated the paper positively. I recommend accept for this paper.